# Antibody affinity maturation and plasma IgA associate with clinical outcome in hospitalized COVID-19 patients

Juanjie Tang[1,3], Supriya Ravichandran[1,3], Youri Lee[1,3], Gabrielle Grubbs[1,3], Elizabeth M. Coyle[1], Laura Klenow[1], Hollie Genser[2], Hana Golding[1] & Surender Khurana [1✉]

Hospitalized COVID-19 patients often present with a large spectrum of clinical symptoms. There is a critical need to better understand the immune responses to SARS-CoV-2 that lead to either resolution or exacerbation of the clinical disease. Here, we examine longitudinal plasma samples from hospitalized COVID-19 patients with differential clinical outcome. We perform immune-repertoire analysis including cytokine, hACE2-receptor inhibition, neutralization titers, antibody epitope repertoire, antibody kinetics, antibody isotype and antibody affinity maturation against the SARS-CoV-2 prefusion spike protein. Fatal cases demonstrate high plasma levels of IL-6, IL-8, TNFα, and MCP-1, and sustained high percentage of IgA-binding antibodies to prefusion spike compared with non-ICU survivors. Disease resolution in non-ICU and ICU patients associates with antibody binding to the receptor binding motif and fusion peptide, and antibody affinity maturation to SARS-CoV-2 prefusion spike protein. Here, we provide insight into the immune parameters associated with clinical disease severity and disease-resolution outcome in hospitalized patients that could inform development of vaccine/therapeutics against COVID-19.

[1] Division of Viral Products, Center for Biologics Evaluation and Research (CBER), FDA, Silver Spring, MD, USA. [2] Quest Diagnostics at Adventist Healthcare, Rockville, MD, USA. [3] These authors contributed equally: Juanjie Tang, Supriya Ravichandran, Youri Lee, Gabrielle Grubbs. ✉email: Surender.Khurana@fda.hhs.gov

As of September 2020, the numbers of SARS CoV-2 infected individuals in the world surpassed 34 million and the rate of mortality is around 4–6% globally[1]. The disease manifestations vary widely, ranging from asymptomatic[2], to mild symptomatic patients who recover at home, to severe cases that require hospitalization. Some hospitalized patients are released after palliative treatments with no admission to the ICU, while others deteriorate rapidly, develop acute respiratory-distress syndrome (ARDS), and are admitted to ICU for more extensive treatments including ventilators[3–6]. Multiple studies are ongoing to better understand the clinical manifestations and the contributions of innate and adaptive immunity to COVID-19 disease outcome[2,7]. Cytokine storm has been associated with severe cases of COVID-19 patients[8]. However, it is still not fully understood whether the severe clinical disease is due to failure of the adaptive immune response to control viral replication and resolve the inflammatory response[9–13]. Therefore, there is need to perform in-depth evaluation of the immune response in hospitalized patients following SARS-CoV-2 infection to identify the immune markers associated with differences between those who survive and those who succumb to the COVID-19 disease[14].

To better understand the immune correlates of disease severity or disease resolution, in this study, we performed comprehensive longitudinal analysis of antibody response in 229 samples collected frequently from hospitalized COVID-19 patients with different clinical outcomes (fatal vs. ICU recovered vs. non-ICU recovered) in two hospitals in Maryland. The plasma samples were evaluated by Genome Fragment Phage Display Library spanning the complete SARS-CoV-2 spike gene (SARS-CoV-2 GFPDL) to determine polyclonal antibody epitope repertoire during patient hospitalization from the day of admission (1–10 days post onset of symptoms) and the last sample available either at death or discharge[15]. Phage display technique is suitable for display of properly folded and conformationally active proteins as it has been widely used for display of large functionally active antibodies, enzymes, hormones, viral and mammalian proteins. We have adapted this technology for unbiased, comprehensive Genome Fragment Phage Display Library (GFPDL) approach for multiple viral pathogens including SARS-CoV-2, Ebola, Zika, Influenza, RSV, HIV, to define antibody epitope repertoire of post-vaccination/infection samples to define both linear and conformation epitopes[16–21].

In addition, we employed several immunoassays including cytokine profiling, SARS-CoV-2 pseudovirion neutralization assay, hACE2-receptor blocking to determine functional activity of the polyclonal immune response. Surface plasmon resonance (SPR)-based real-time kinetics assay was used to measure antibody binding, isotype-class switching and antibody affinity maturation against the prefusion spike protein, which reflects the quality of the polyclonal immune response elicited in hospitalized COVID-19 patients[15,22]. We compared the immunological parameters of patients who were discharged from hospital with no ICU admission, with patients who were admitted to ICU and either survived or expired due to COVID-19. Here, we identified sustained high IgA responses and minimal affinity maturation against the SARS-CoV-2 prefusion spike as key immunological parameters of ICU patients that succumb to COVID-19. Our data shows that disease resolution in COVID-19 patients was associated with strong antibody affinity maturation to SARS-CoV-2 prefusion spike protein.

## Results

### Longitudinal analysis of blood cytokines in cohorts of hospitalized patients.
Two-hundred twenty nine sequential plasma samples were collected frequently from 25 acutely hospitalized COVID-19 patients until death, or discharge from the hospital following recovery from disease. Samples were numbered from the day (D) of symptom onset and were all PCR-confirmed. All experimental analyses were performed in a blinded manner. The demographic and clinical information was unblinded for the authors by the hospital staff following data analysis and is presented in Source Data. After unblinding, the patients fell into three groups: (i) moderate patients released from hospital with no ICU admission (Non-ICU survived; NS series; $n = 6$); (ii) severe patients that were admitted to ICU but recovered from symptoms and were discharged from the hospital (ICU-Survived; IS series; $n = 11$); (iii) and COVID-19 patients in ICU who expired (Expired; E series; $n = 8$). Power analysis for sample size is presented in Suppl. Table 1 and Suppl. Equation 1. Distribution of COVID-19 patients and samples used in the study are shown in Suppl. Fig. 1. Most patients (80%) were symptomatic males between the ages of 44 and 75 years.

Cytokine (17-plex) profiling were conducted on all patients (Source Data). Prolonged elevation of multiple cytokines (IL-6, IL-8, TNFα, MCP-1, and MIP-1β) were observed in patients that were admitted to the ICU who either expired (red curves; Fig. 1a–e) or survived (blue curves; Fig.1f–j), compared with patients who were discharged from hospital without ICU admission (green curves; Fig. 1k–o). The area under the curve (AUC) were calculated for the first 20 days of hospitalization for these three cohorts to control for the duration of the samples collected so they were not impacted the window of samples collected. The differences in AUC between ICU vs. non-ICU patients reached statistical significance for IL-6 and IL-8 (Source Data and Fig. 1p). Serum samples from 15 normal healthy adults collected in 2010 were found to contain <10 pg/mL of either IL-6 or IL-8. Cytokine upregulation has previously been shown to be associated with severe cases of COVID-19 patients[8].

### Development of neutralizing and hACE2-receptor blocking antibodies in hospitalized patients with different clinical outcomes.
The majority of the hospitalized patients developed hACE2-blocking antibodies (black curves, Fig. 2), as well as neutralizing antibodies measured in a pseudovirion-neutralization assay (PsVNA50). In general, there was good agreement between the two assays (Suppl. Fig. 2), except for a few patients who retained high ACE2-blocking titers but showed low neutralizing titers at later days of hospitalization in two expired patients (E-58 and E-61) and two patients that survived after ICU admission (IS-55 and IS-77). Interestingly, not all hospitalized patients developed strong neutralizing titers. In each of the groups we identified individuals with either high or low titers, demonstrating discordance between neutralization titers and disease severity and outcome (Fig. 2a–d).

### Epitope repertoire of antibodies in hospitalized COVID-19 patients using SARS-CoV-2 spike Genome Phage Display Library (SARS-CoV-2-S GFPDL).
To understand the development of IgM, IgG and IgA repertoires following SARS-CoV-2 infection during hospitalization, a highly diverse SARS-CoV-2-S GFPDL with $>10^{7.1}$ unique phage clones was used to evaluate plasma samples collected at early and late hospitalization time points from the same patients. The SARS-CoV-2 GFPDL displays epitopes of 18-500 amino acid residues with random distribution of size and sequences of inserts that spans across the SARS-CoV-2 spike gene. Recently, we demonstrated that the SARS-CoV-2-S GFPDL expresses both linear and conformational epitopes, including neutralization targets, that were recognized by post-vaccination sera[15]. GFPDL-based epitope mapping of monoclonal antibodies (MAbs) targeting SARS-CoV-2 spike or RBD

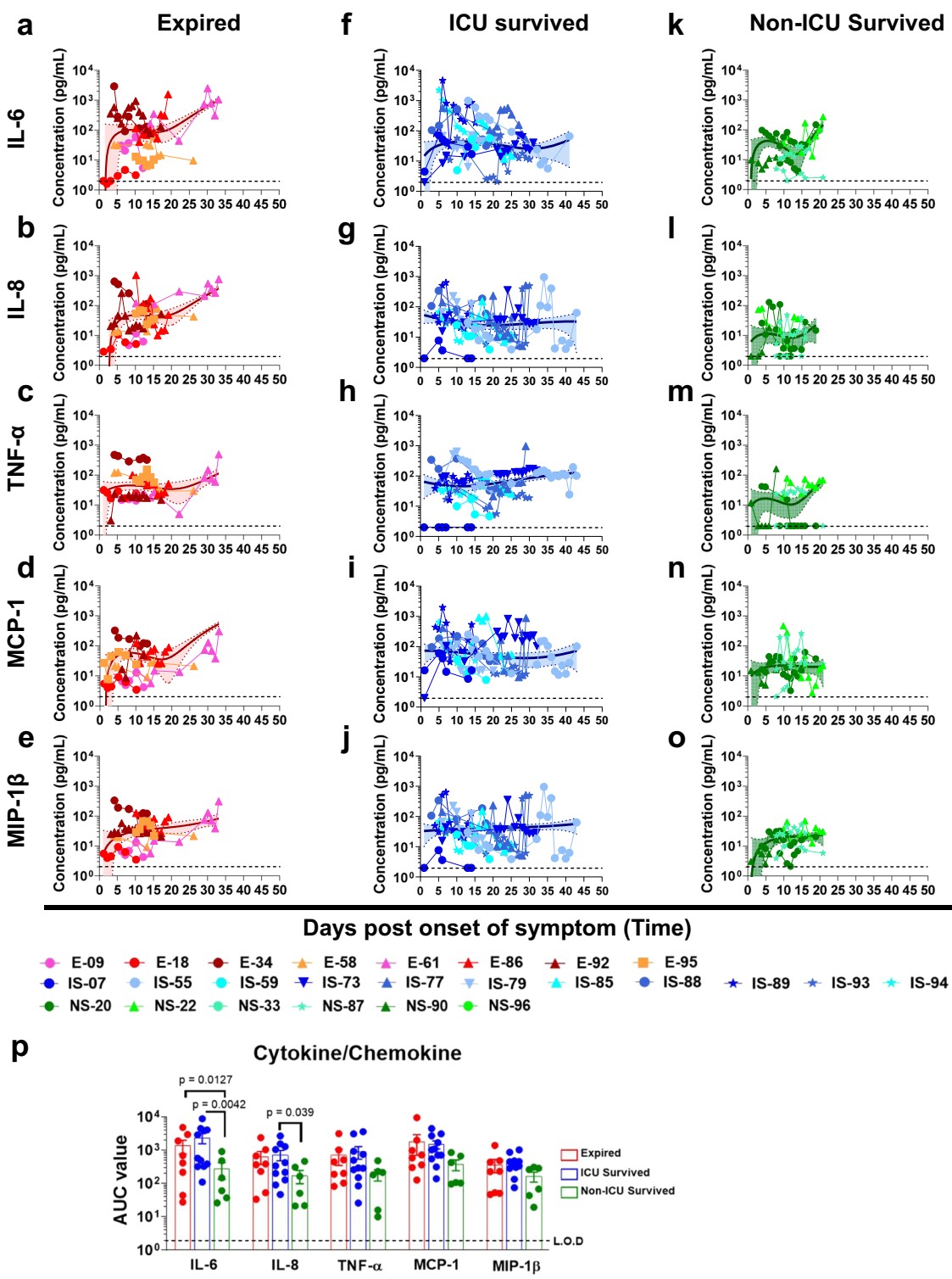

identified the expected epitope recognized by these MAbs. Moreover, in the current study, the SARS-CoV-2 GFPDL adsorbed majority (>91%) of SARS-CoV-2 prefusion spike-specific antibodies in the post-infection polyclonal human plasma samples either from expired or survived patients, providing proof-of-concept for use of the SARS-CoV-2 GFPDL for epitope repertoire analyses of human plasma (Suppl. Fig. 3). Sera from uninfected individuals (collected in 2009) bound very few (<10) phages of the SARS-CoV-2 GFPDL.

The post-infection polyclonal IgM, IgA and IgG antibody-epitope repertoires were determined by GFPDL analysis conducted

on polyclonal samples pooled from the first available time-point (within 1–4 days post-onset of symptoms) vs. plasma from the last time-point of either fatal COVID-19 patients (E-18, E-34, and E-58; Expired—<D4 & Expired—final sample) (shown in red in Fig. 3b, c, d), or non-ICU, surviving COVID-19 patients (NS-33 and NS-90; Survived—<D4 & Survived—Discharged) (shown in green in Fig. 3b, c, d). The total number of bound phages by IgM, IgG, and IgA antibodies from each patient group is shown in Fig. 3a. On the day of hospitalization, the numbers of IgM, IgG, and IgA bound phages ranged between $10^4$–$10^5$ in both groups. The number of phages bound by the pooled samples from the latest time points

**Fig. 1 Cytokine/chemokine analyses of COVID-19 patients during hospitalization.** Cytokine/chemokine levels in COVID-19 patients at different days post-onset of symptoms. (**a-e**); Expired patients ($n = 8$), (**f-j**); Survived patients (ICU; $n = 11$), (**k-o**); Survived patients (non-ICU; $n = 6$). IL-6 (**a**, **f**, and **k**), IL-8 (**b**, **g**, and **l**), TNFα (**c**, **h**, and **m**), MCP-1 (**d**, **i**, and **n**), and MIP-1β (**e**, **j**, and **o**) concentrations in 4-fold diluted plasma samples at various time-points after hospitalization of the COVID-19 patients were determined via a Bio-Plex Pro Human Cytokine Panel 17-Plex assay. **a-o** Expired patients (patient number's starting with E); ICU-survived patients (patient number's starting with IS); and non-ICU survived patients (patient number's starting with NS). The trendline fits were performed for expired (red), ICU survivors (blue) and non-ICU survivors (green) using a non-linear regression model with polynomial distribution through the origin in Graphpad Prism. The trend line is depicted as solid colored line with the error bands representing 95% confidence interval shown as shaded colored area for each group. **p** Area under the curve (AUC) was determined for the five cytokine/chemokines from day 1 to day 20 of the three COVID-19 patient groups to control for the window of samples collected. Bar chart shows datapoints for each individual and presented as mean values ± SEM. The statistical significances between the groups of area under curve (AUC values) for "expired" patients (**a-e**; shades of red; $n = 8$ biologically independent individuals), "ICU-survived" patients (**f-j**; shades of blue; $n = 11$ biologically independent individuals), and "non-ICU survived" patients (**k-o**; shades of green; $n = 6$ biologically independent individuals) were determined by non-parametric (Kruskal–Wallis) statistical test using Dunn's multiple comparisons analysis in GraphPad prism. The differences were considered statistically significant with a 95% confidence interval when the $p$-value was <0.05.

increased by ~10-fold for IgG in both groups. However, the number of phages bound to IgM and IgA at the latest time point decreased in the survivors but increased in the fatal COVID-19 patients (Fig. 3a). The total serum antibodies in these patients were in the same range as has been observed for normal adults (IgG: mean 9.6 (range 7.1–12.9) mg/mL, IgA: mean 2.8 (range 1.7–3.4) mg/mL, and IgM: mean 1.9 (range 1.1–3.8) mg/mL).

The IgM-epitope repertoire was diverse and covered the entire spike protein including the receptor-binding domain (RBD) and the receptor-binding motif (RBM) in S1, and the fusion peptide in S2. No obvious difference were found in the epitope diversity of IgM antibodies between the two groups (Fig. 3b red vs. green). The IgG epitope recognition was also diverse, with immunodominant antibody epitopes at the N-terminal domain (NTD) and C-terminus of S1 domain, N-terminus of and β-rich connector domain (CD) and the HR2 domain at the C-terminus of S2, followed by HR1 domain of S2 in both the fatal cases and survived patients (Fig. 3c). Importantly, in the expired group, we observed minimal recognition of a short epitope sequences in RBD, and no binding to receptor binding motif (RBM) (Fig. 3c). The IgG antibodies in the non-ICU survived pooled plasma recognized the entire RBM region, and also bound to an immunodominant epitope in the fusion peptide of S2 domain. Interestingly, the IgA repertoire of the expired patients also had a "hole" in the RBD/RBM region (similar to the IgG repertoire of the same plasma pool, which was not seen with the pooled plasma from the surviving (non-ICU) patients (Fig. 3d).

Subsequently, we conducted IgM, IgG and IgA GFPDL analysis on pooled samples from ICU- admitted patients who survived (IS-07, IS-88, and IS-89) (Fig. S4). The IgG/IgA antibodies in these ICU-survivors did recognize the entire RBM region and fusion peptide in the S2 domain, similar to the non-ICU surviving patients (Suppl. Fig. 4 vs. Fig. 3c, d).

Overall, SARS-CoV-2 infection in these COVID-19 patients generated a diverse antibody repertoire across the spike protein. These included twelve antigenic regions and thirteen antigenic sites within these regions and were defined by at least 4% of phage clones obtained after affinity selection on IgM/IgG/IgA antibodies with at least one plasma sample at any time point (Fig. 4a, b). Most regions/sites (length of 34 to 239 amino acid residues) were similarly recognized by antibodies in all patient groups (shown in black in Fig. 4a, b). However, comparison of the IgG epitope repertoire between the expired vs. survived COVID-19 patients revealed that surviving patients contained IgG and IgA antibodies that bound to the large RBD site S6 encompassing the RBM (shown in green in Fig. 4a, b) and that was missed by the IgG as well as IgA antibodies from expired patients. In addition, the fusion peptide containing antigenic region S9 in the S2 domain (in green) was primarily recognized by IgG from surviving

patients but not by IgG in fatal COVID-19 cases. Structural depiction of these antigenic sites on the SARS-CoV-2 spike (on PDB#6VSB) demonstrated that both antigenic site S6 (Fig. 4c) and site S9.3 (Fig. 4d) are surface-exposed on the trimeric spike[23]. These antigenic regions/sites in the S1 domain are not conserved among other coronaviruses. However, some sites in S2 show >50% sequence conservation across multiple human and bat CoV species (Source Data).

To further evaluate the specificity of post-SARS-CoV-2 infection antibodies in serum to these two differential immunodominant antigenic sites in spike that were not bound at high frequency in IgG/IgA from COVID-19 expired patients using GFPDL analysis, these peptides were chemically synthesized and analyzed in ELISA (Fig. 4e, f). Using either the early or last plasma from survivors showed higher plasma IgG binding to the RBM peptide (Fig. 4e), that reached statistical significance in comparison between fatal vs. non-ICU survivors. On the other hand, binding to the fusion peptide (Fig. 4f) increased between early and late time points for the surviving patients (shown in blue and green), but not for expired patients, and reached statistical significance.

Therefore, the GFPDL analysis identified potential holes in the antibody repertoires of expired patients. Specifically, in the late stage of the disease, there was reduced RBM-and Fusion-peptide binding by IgG and IgA antibodies from expired patients compared with the patients who survived COVID-19.

**Evolution of antibody binding and isotype class-switching against SARS-CoV-2 prefusion spike following SARS-CoV-2 infection in hospitalized COVID-19 patients.** To determine the evolution of antibody response in these 25 COVID-19 patients during their hospital stay, quantitative and qualitative SPR analyses was performed for several dilutions (10-, 50-, and 250-fold) of polyclonal plasma from all time-points from each patient using recombinant purified SARS-CoV-2 prefusion spike protein. The conformational integrity of the prefusion spike used in our SPR assay was assessed using recombinant human ACE2 (hACE2), the SARS-CoV-2 receptor, that demonstrated a high-affinity interaction with an affinity constant of 6.7 nM (Suppl. Fig. 5). Representative sensorgrams of binding to the prefusion spike by serially diluted plasma (10-, 50- and 250-fold) from two patients are shown in Suppl. Fig. 6.

Control human plasma samples collected from healthy adults in 2008 showed <6 RU binding to the SARS-CoV-2 prefusion spike protein in SPR (Suppl. Figs. 6 and 7). We previously demonstrated that the optimized SPR does not show non-specific background reactivity against the SARS-CoV-2 spike proteins with plasma from unimmunized rabbits or rabbits immunized with an irrelevant antigen[15].

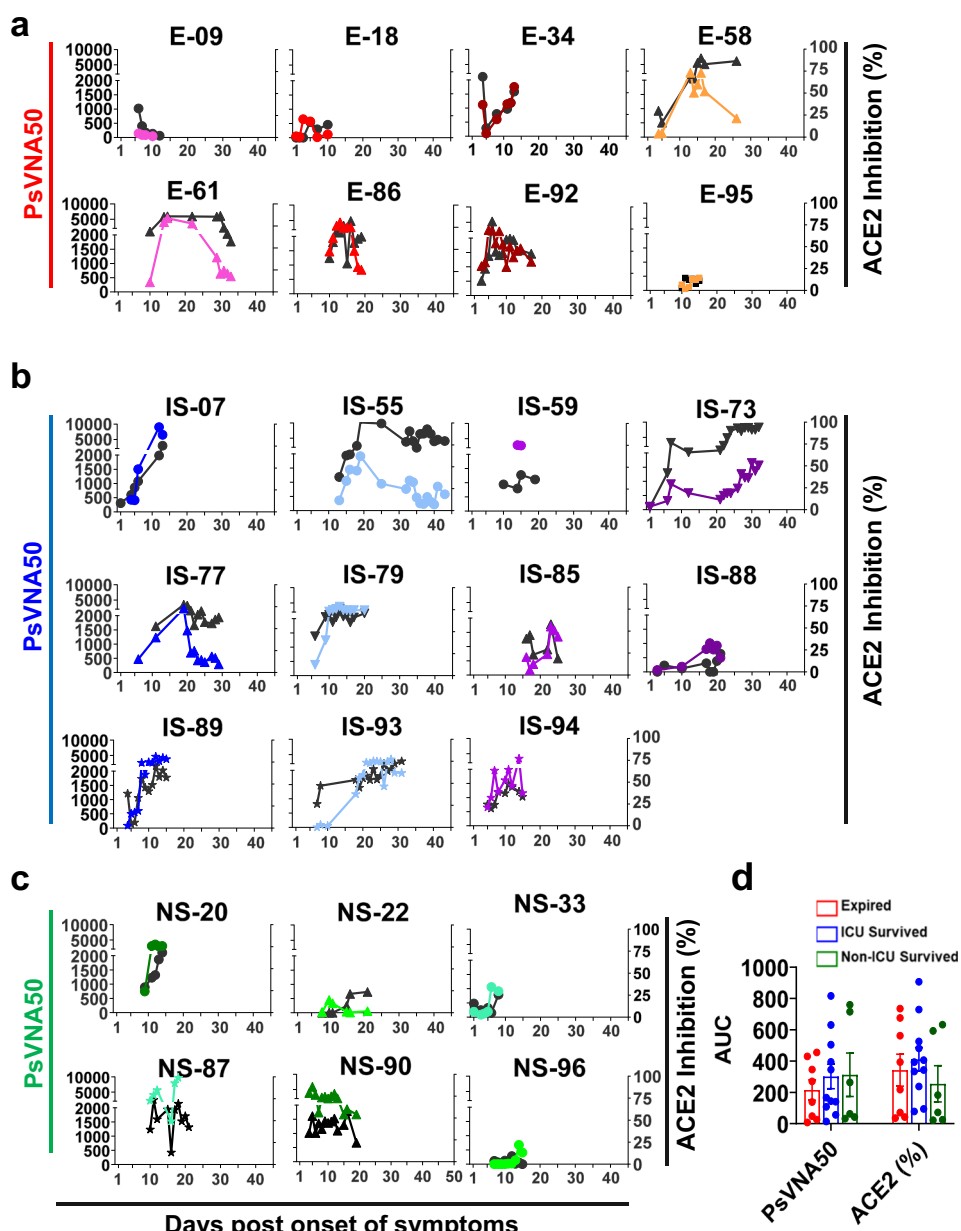

**Fig. 2 Neutralizing antibody titers and hACE2 receptor inhibition activity of COVID-19 patients' plasma during hospitalization.** SARS-CoV-2 neutralizing antibody titers (PsVNA50) in plasma at various time-points as determined by pseudovirion neutralization titer 50 (PsVNA50) in Vero E6 cells are shown in colored curves, and the percentage of ACE2 inhibition values at different days post-onset of symptoms are shown in black curves. **a** "Expired" patients ($n = 8$); **b** "ICU-survived" patients ($n = 11$); and **c** "non-ICU survived" ($n = 6$) patients. Expired patients (patient number's starting with e); ICU-survived patients (patient number's starting with IS); and non-ICU survived patients (patient number's starting with NS). Virus neutralization PsVNA50s titers were calculated with GraphPad prism version 8. Percent inhibition of hACE2 binding to RBD in presence of 1:100 dilution of COVID-19 plasma was measured by ELISA. The neutralization and ACE2-inhibition experiments were performed twice independently with similar results. **d** Area under the curve (AUC) was determined for the PsVNA50 or the percent ACE2 inhibition from day 1 through day 20 of the three COVID-19 patient groups to control for the window of samples collected. Bar chart shows datapoints for each individual and presented as mean values ± SEM. The statistical significances between the groups were determined by non-parametric (Kruskal–Wallis) statistical test using Dunn's multiple comparisons analysis in GraphPad prism for the area under curve (AUC values) between "expired" patients (red; $n = 8$ biologically independent individuals), "ICU-survived" patients (blue; $n = 11$ biologically independent individuals), and "non-ICU survived" patients (green; $n = 6$ biologically independent individuals) did not identify any statistical significance ($p > 0.05$).

In addition to measuring bound antibodies specific for the prefusion spike protein, the SPR assay was used to determine the relative contribution of each antibody isotype: IgM, IgG (including subclasses) and IgA in plasma antibody bound to prefusion spike in these COVID-19 patients (Fig. 5).

In the patients that were discharged with no ICU admission (NS series), the total antibody IgG binding increased from the day of hospitalization to the day of discharge. During that time, the IgM antibodies to prefusion spike were high on days 7–10 post-onset of symptoms and then class-switched with increasing contributions of primarily IgG1, followed by IgG2, with minimal IgG3 and IgG4. A small IgA component was measured in some of these patients that was <10% at most-time points during hospitalization (Fig. 5c).

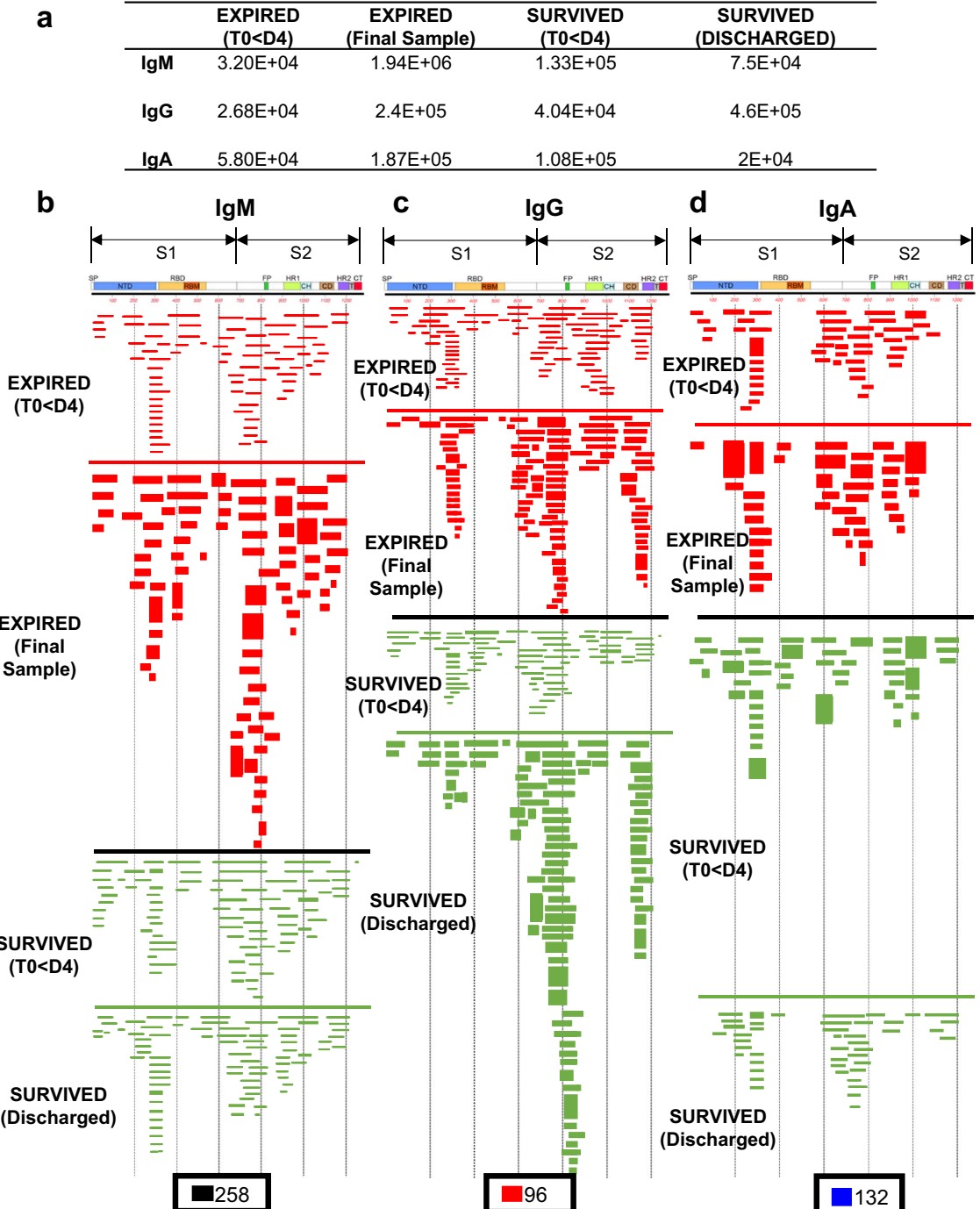

**a**

| | EXPIRED (T0<D4) | EXPIRED (Final Sample) | SURVIVED (T0<D4) | SURVIVED (DISCHARGED) |
|---|---|---|---|---|
| IgM | 3.20E+04 | 1.94E+06 | 1.33E+05 | 7.5E+04 |
| IgG | 2.68E+04 | 2.4E+05 | 4.04E+04 | 4.6E+05 |
| IgA | 5.80E+04 | 1.87E+05 | 1.08E+05 | 2E+04 |

**Fig. 3 IgM, IgG, and IgA antibody epitope repertoires elicited in expired vs. survived (non-ICU) hospitalized COVID-19 patients.** Distribution of phage clones after affinity selection on plasma samples collected at early vs. late time points from hospitalized patients. **a** Number of IgM, IgG, and IgA bound phage clones selected using SARS-CoV-2 spike GFPDL on pooled polyclonal samples from "expired" patients (E-18, E-34, and E-58), collected from days 1–4 (Expired—<D4) following symptom onset and the last day before death (Expired— Final sample) or pooled polyclonal samples from "survived" non-ICU COVID-19 patients (NS-33 and NS-90) collected on days 1–4 (Survived—<D4) following symptom onset and the day of discharge (Survived—Discharged) from hospital. **b–d** IgM, IgG, and IgA antibody epitope repertoires of expired (red) vs. survived (green) COVID-19 patients and their alignment to the spike protein of SARS-CoV-2. Graphical distribution of representative clones with a frequency of ≥2, obtained after affinity selection, are shown. The horizontal position and the length of the bars indicate the alignment of peptide sequence displayed on the selected phage clone to its homologous sequence in the SARS-CoV-2 spike. The thickness of each bar represents the frequency of repetitively isolated phage. Scale value for IgM (black), IgG (red), and IgA (blue) is shown enclosed in a black box beneath the respective alignments. The GFPDL affinity selection data was performed in duplicate (two independent experiments by researcher in the lab, who was blinded to sample identity), and similar number of phage clones and epitope repertoire was observed in both phage display analysis.

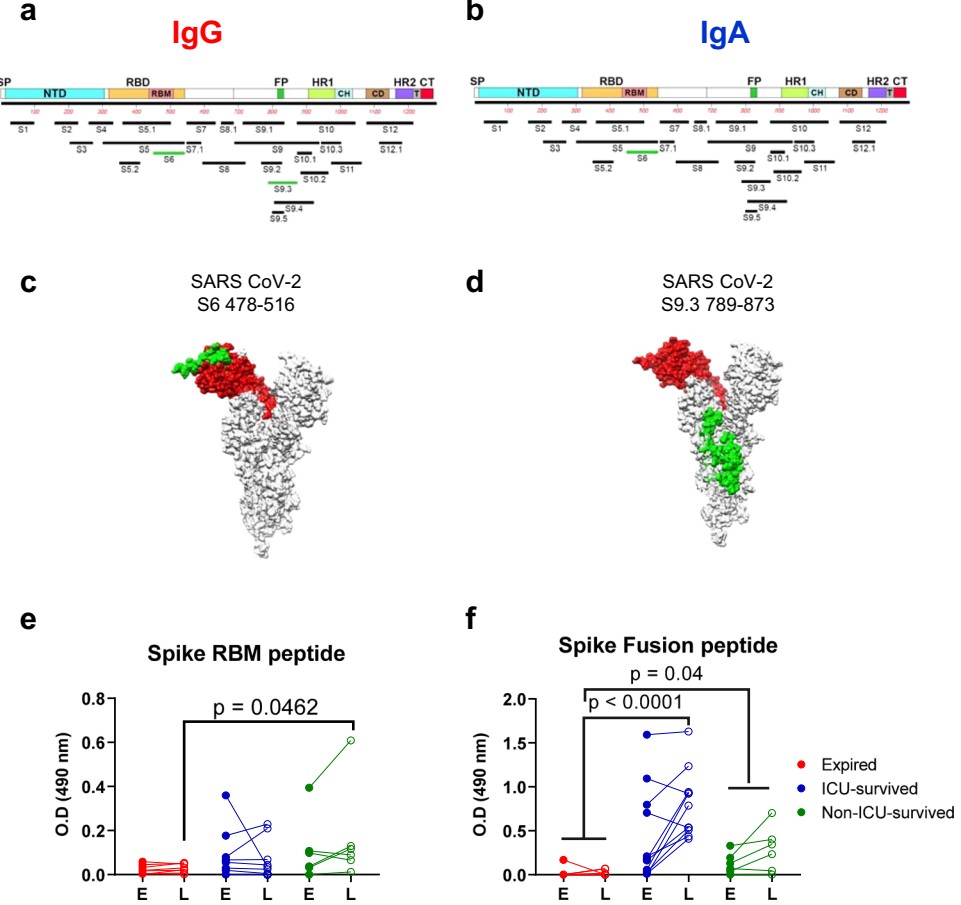

**Fig. 4 Antibody epitope profile against the Spike protein following SARS-CoV-2 infection.** Antigenic regions/sites within the spike protein recognized by plasma antibodies following SARS-CoV-2 infection (based on data presented in Fig. 3). The amino acid designation is based on the spike protein sequence encoded by the SARS-CoV-2 Wuhan-Hu-1 strain (GenBank: MN908947.3) spike. The antigenic regions/sites discovered using the post-infection antibodies are depicted below the SARS-CoV-2 spike schematic. The epitopes of each protein are numbered in a sequential fashion indicated in black. Antigenic sites shown in green letters (SARS CoV-2 S6 and S9.3) were uniquely recognized by post-SARS-CoV-2 infection IgG (**a**) or IgA (**b**) antibodies only in the "survived" but not from expired COVID-19 patients (as shown in Fig. 3). **c**, **d** Structural representation of S6 (**c**) and S9.3 (**d**) antigenic sites depicted in green on the surface of a monomer in a trimeric spike (PDB#6VSB) with a single receptor-binding domain (RBD) in the up conformation, wherever available using UCSF Chimera software version 1.11.2. The RBD region is shaded in red (residues 319–541) on both structures. **e**, **f** Seroreactivity of COVID-19 plasma samples with the selected synthetic peptides covering epitope S6 (receptor-binding motif; RBM) and peptide S9.3 (fusion peptide; FP). Absorbance at 100-fold dilution of early first sample (E) or last (L) sample from SARS-CoV-2 infected individuals were tested for binding to RBM (panel **e**) and FP (panel **f**) in IgG ELISA. The statistical significances between "expired" patients (red; $n = 8$ biologically independent individuals), "ICU-survived" patients (blue; $n = 11$ biologically independent individuals), and "non-ICU survived" patient (green; $n = 6$ biologically independent individuals) groups were determined by non-parametric (Kruskal–Wallis) statistical test using Dunn's multiple comparisons analysis in GraphPad prism. $p$-values <0.05 were considered significant with a 95% confidence interval.

The binding of antibodies to prefusion spike protein in half of the ICU patients who expired (Fig. 5a) or survived (Fig. 5b) showed binding curves, with initial increase in total antibody binding followed by a decrease in RU values (Suppl. Fig. 8a). The initial increase in spike-binding antibody titers were faster in the more severe cases (ICU admitted) compared with the non-ICU cases (Suppl. Fig. 8a). The overall prefusion spike binding antibodies were significantly higher in severe ICU cases compared with non-ICU patients (Fig. 5d).

The IgM responses trended higher in the non-ICU patients (Fig. 5e). In the case of IgG responses, most of the ICU patients (14 of the 19) contained >10% prefusion spike-binding IgG antibodies that comprised either IgG3 or IgG4 subclass, in addition to IgG1 response. Importantly, the isotype distribution demonstrated sustained high percentage of IgA binding antibodies in all ICU patients, either expired or survived (Fig. 5a, b), not seen in the non-ICU patients (Fig. 5c). The percentages of

IgA antibodies bound to prefusion spike were significantly higher in ICU patients (either expired or survived) compared with the non-ICU patients (Fig. 5e).

**Antibody affinity maturation during hospitalization comparing fatal vs. survived COVID-19 patients.** In addition to total binding antibodies, it was important to determine if SARS-CoV-2 infection induced antibody affinity maturation against the viral spike protein. Technically, since antibodies are bivalent, the proper term for their binding to multivalent antigens like viruses is avidity, but here we use the term affinity throughout, since we measured primarily monovalent interactions[15,20,22]. Serial dilutions (10-, 50-, and 250-fold) of freshly prepared plasma were analyzed for antibody kinetics on SARS-CoV-2 prefusion spike captured via His-tag on the chip at low surface density to measure monovalent interactions independent of the antibody isotype. Antibody off-rate

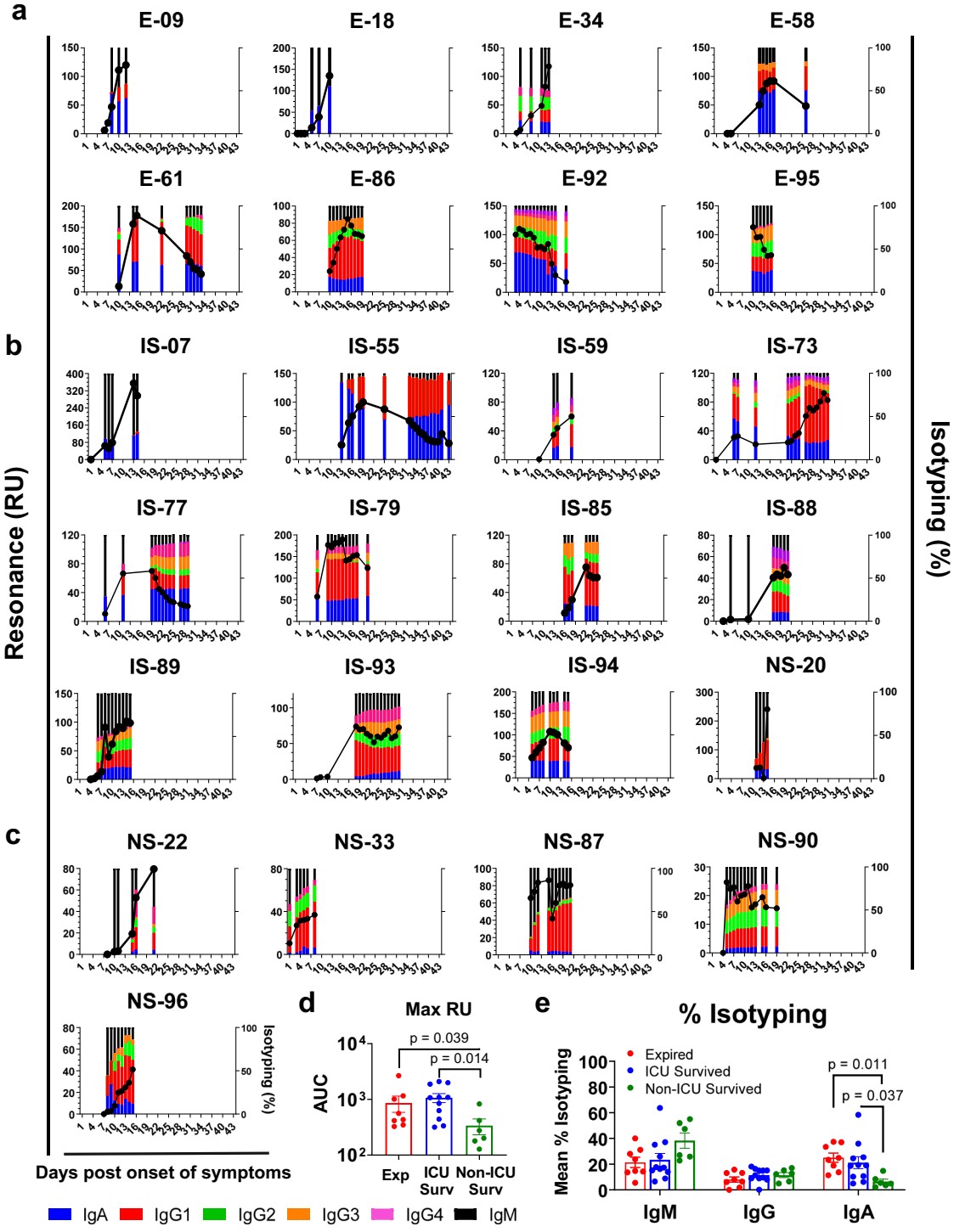

**Days post onset of symptoms**

IgA IgG1 IgG2 IgG3 IgG4 IgM

constants, which describe the stability of the antigen-antibody complex, i.e., the fraction of complexes that decays per second in the dissociation phase, were determined directly from the human polyclonal plasma sample interaction with recombinant purified SARS CoV-2 prefusion spike ectodomain using only sensorgrams with Max RU in the range of 10–100 RU. Off-rate constants were determined from two independent SPR runs.

In the expired COVID-19 patients, low antibody affinity (faster antibody dissociation kinetics) was observed to the prefusion spike with mean final day off-rate of 0.0173 per second (range of 0.045 to 0.0056 per second) during hospitalization (Fig. 6a and Suppl. Fig. 8b, red curve). In these fatal cases, minimal or no anti-

spike antibody affinity maturation was observed from the initial sample to the last sample until their demise. Patients that survived after ICU admission demonstrated a gradual increase in antibody affinity over time with a mean final day off-rate of 0.0023 per second (range of 0.0071 to 0.0012 per second) (Fig. 6b and Suppl. Fig. 8b, blue curve). Importantly, the non-ICU patients demonstrated even higher antibody affinity maturation (2-log slower dissociation rates) with mean final day off-rates reaching 0.0012 per second (ranging between 0.0015 to 0.00096 per second) prior to their discharge (Fig. 6c and Suppl. Fig. 8b, green curve). The anti-spike antibody affinities (as measured by off-rates) were significantly higher in survivors (ICU or non-ICU)

**Fig. 5 Evolution of antibody binding and antibody isotype following SARS-CoV-2 infection in COVID-19 patients and association with disease severity.**
Serial dilutions of each plasma sample collected at different time points from the COVID-19 patients were analyzed for antibody binding to purified SARS-CoV-2 prefusion spike ectodomain (aa 16-1213) lacking the cytoplasmic and transmembrane domains (delta CT-TM), and containing His tag at C-terminus, was produced in FreeStyle293-F mammalian cells. Total antibody binding is represented as SPR maximum resonance units (RU) (black curves) of 10-fold diluted plasma samples from expired patients (**a**; patient number's starting with E; $n = 8$), ICU-surviving patients (**b**; patient number's starting with IS; $n = 11$) and non-ICU surviving patients (**c**; patient number's starting with NS; $n = 6$). Isotype composition of plasma antibodies bound to SARS-CoV-2 spike prefusion protein for each individual COVID-19 patient at different time-points as measured in SPR. The resonance units for each antibody isotype was divided by the total resonance units for all the antibody isotypes combined to calculate the percentage of each antibody isotype (according to the color codes; IgM, black; IgA, blue; IgG1, red; IgG2 green; IgG3, orange; IgG4, fuchsia). All SPR experiments were performed twice blindly. The variation for each sample in duplicate SPR runs was <5%. **d** The area under the curve (AUC) of SARS-CoV-2 prefusion spike binding antibody levels (Max RU) for the COVID-19 patients who expired (red; $n = 8$ biologically independent individuals) vs. ICU-survived (blue; $n = 11$ biologically independent individuals) vs. non-ICU survived (green; $n = 6$ biologically independent individuals). Bar chart shows datapoints for each individual and presented as mean values ± SEM. **e** AUC of mean percentages of antibody isotypes IgM, IgG, IgA bound to SARS-CoV-2 prefusion spike for the COVID-19 patients who expired (red; $n = 8$ biologically independent individuals) vs. ICU-survived (blue; $n = 11$ biologically independent individuals) vs. non-ICU survived (green; $n = 6$ biologically independent individuals). Bar chart shows datapoints for each individual and presented as mean values ± SEM. The statistical significances between the groups were determined by non-parametric (Kruskal–Wallis) statistical test using Dunn's multiple comparisons analysis in GraphPad prism. The differences were considered statistically significant with a 95% confidence interval when the p-value was <0.05.

compared with the fatal COVID-19 patients (Fig. 6d). In agreement, significantly higher antibody affinity maturation (fold change in antibody off-rate of last time-point from first-time point during hospitalization) was observed in survivors (ICU or non-ICU) compared with fatal cases, with mean fold change in antibody dissociation rate of 61.1 for non-ICU survivor's vs. 22.5 for ICU survivor's vs. 1.97 for fatal cases (Fig. 6e).

## Discussion
Differences in antibody kinetics between COVID-19 patients with different disease severity have been reported, including virus neutralization titers and spike-binding antibodies following acute infection or during convalescence[24]. However, the predictive value of neutralization titers with disease outcome has not been established, since severe COVID-19 patients show much faster and stronger neutralizing antibodies compared with mild cases. Therefore, an immune marker that correlates with protection is not well-defined and are critically needed to inform COVID-19 medical countermeasures during ongoing pandemic.

To explore antibody markers that may correlate with disease severity or resolution of disease caused by SARS-CoV-2 infection, we performed a comprehensive longitudinal antibody analysis on SARS-CoV-2 PCR-confirmed hospitalized COVID-19 patients. They included individuals who were discharged from hospital with no ICU-admission, as well as more severe patients who required ICU admission, of whom some survived, and while a few met a worse outcome and succumbed to the disease. Importantly, we were able to obtain sequential samples on all hospitalized patients (total 229 plasma). In all plasma samples we measured cytokines levels, antibody neutralization titers, epitope repertoire by GFPDL, isotype diversity, and antibody affinity maturation by SPR. In spite of the moderate group sizes, we had the statistical power to identify significant differences between the groups.

As expected, the more severe COVID-19 patients demonstrated sustained high levels of proinflammatory cytokines/chemokines compared with the non-ICU patients, that reached statistical significance for IL-6 and IL-8, as reported by others[8–12]. The neutralizing antibody titers varied among patients in all three groups and did not predict disease outcome.

The SARS-CoV-2 GFPDL analyses with plasma from severe patients revealed diverse IgM, IgG, and IgA antibody epitope repertoires. While IgM repertoires were indistinguishable among the cohorts, only in the expired group we identified "holes" in the IgG and IgA antibody repertoires, with epitopes mapping to the receptor binding motif (RBM) in S1 and the fusion peptide in S2, which was confirmed by peptide ELISA. One of the possible

limitations of GFPDL-based assessments is that while the phage display is likely to detect both conformational and linear epitopes on spike protein, they are unlikely to detect paratopic interactions that require post-translational modifications and rare quaternary epitopes that cross-protomers. However, in the in the current study (Fig. S3) and a prior study with post-spike vaccination serum polyclonal antibodies, >91% of anti-spike antibodies were removed by adsorption with the SARS-CoV-2-GFPDL, supporting the use of the GFPDL for analyses of human sera.

The total binding of antibodies to the SARS-CoV-2 prefusion spike protein revealed daily increases throughout the hospitalization period with contributions from IgM, IgA, and IgG isotypes. The kinetics of increase in antibody binding (and virus-neutralization titers) were faster for the more severe COVID-19 patients (deceased or survived), in agreement with recent study[25]. Additionally, the relative contribution of IgA to spike binding antibodies was higher and longer-lasting in most of the severe patients compared with the non-ICU patients throughout the hospital stay. The role of plasma IgA antibodies on SARS-CoV-2 is not clear, since these antibodies may not reach the site of virus infection in the upper and lower respiratory tract. IgA also lacks the effector function of IgG in ADCC and complement activation. A recent study suggested that systemic IgA antibody binding to memory B cells has a negative regulatory activity on antigen receptor B-cell activation[26].

Antibody affinity maturation of prefusion spike-binding antibodies longitudinally during the hospital stay provided a clear antibody correlate differentiating between the survivors vs. fatal cases of COVID-19 disease. Despite the similar anti-spike total antibody-binding titers, we found that antibodies from expired patients had significantly lower affinity maturation over the 1–4 weeks hospitalization compared with 22 to 61-fold mean antibody affinity maturation in COVID-19 survivors. Our findings could be explained by the recent report that found loss of Bcl6-expressing T follicular cells (Tfh) and germinal centers (GC) in autopsy of respiratory tissues (including lymph nodes) from deceased COVID-19 patients[27]. Therefore, it seems that in more severe COVID-19 patients, even though they can generate high binding and neutralizing antibody titres, there is a block to antibody affinity maturation that may be linked to deficiency in CD4 cells, and especially T-follicular helper cells subsets, which are required for entry into germinal center (GC).

In earlier studies with vaccines against H7N9 avian influenza, we found a significant correlation between antibody affinity against the hemagglutinin HA1 globular domain and control of lung virus loads after challenge of ferrets with H7N9[28]. In a study of patients recovering from Zika virus (ZIKV) infections, higher

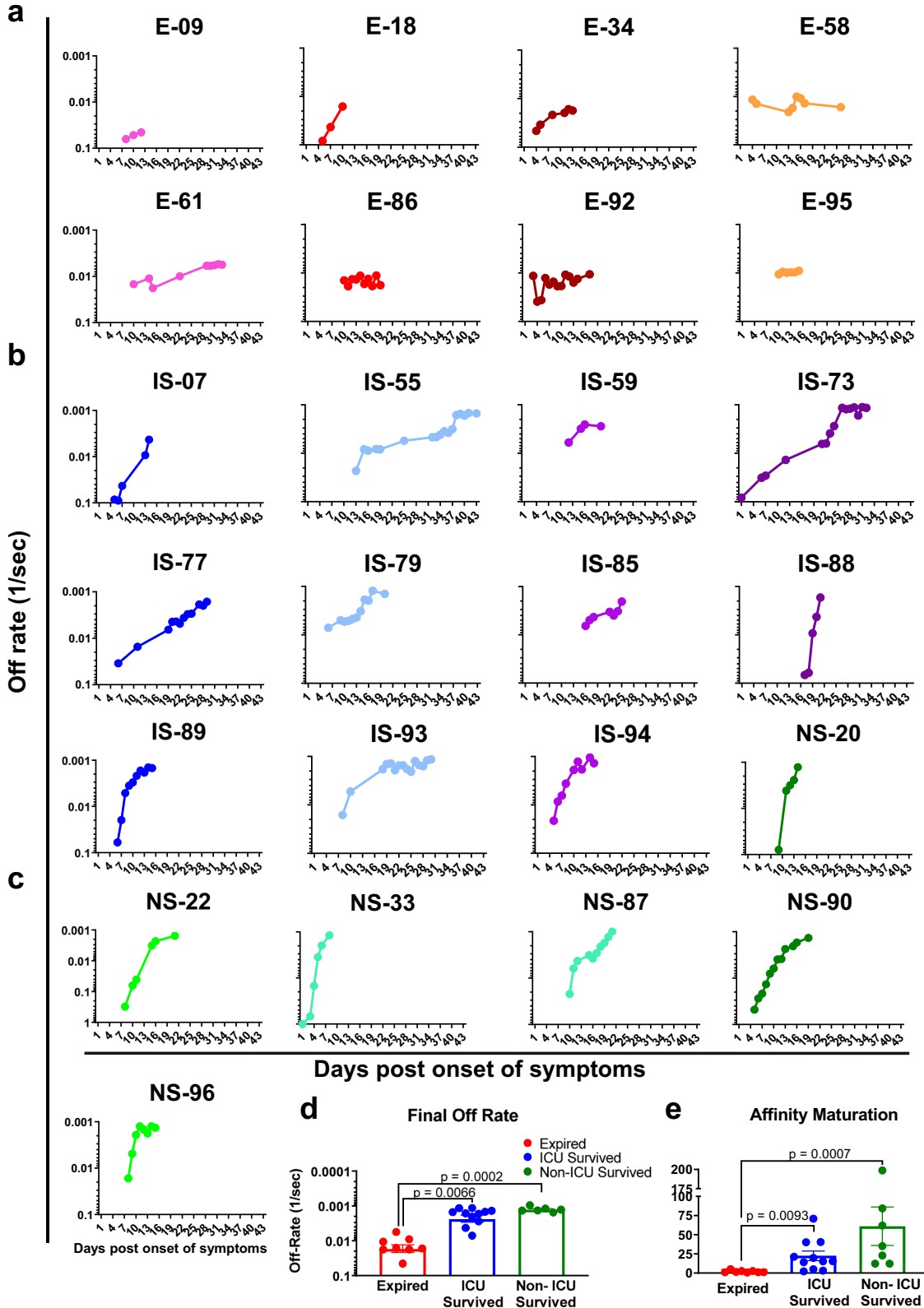

antibody affinity against ZIKV E-DIII correlated with lower clinical scores[21]. A recent longitudinal study of affinity maturation in a survivor of Ebola virus showed that increased affinity to Ebola virus GP was associated with a rapid decline in viral replication and illness severity in this patient[20]. Thus, the ability of virus-specific B cells to enter germinal centres (GC) in lymph nodes and undergo affinity maturation may have consequence on

the ultimate effectiveness of viral control and clearance. This is particularly important with emerging infectious diseases in naïve populations. Indeed, it was reported that many antibodies generated in response to infection with SARS CoV-2 use germline VH and VL genes with limited somatic mutations[29,30]. Additionally, multiple studies described complex immune dysregulation in severe COVID-19 patients that included CD4 cytopenia,

**Fig. 6 Antibody affinity maturation of human antibody response following SARS-CoV-2 infection in COVID-19 patients and association with clinical outcome. a–c** Polyclonal antibody affinity to SARS-CoV-2 prefusion spike protein for COVID-19 patients at different time-points post-onset of symptoms was determined by SPR. Binding affinity was determined for individual COVID-19 patients, **a** expired (in red shades; patient number's starting with E; $n =$ 8); **b** ICU-survived (in blue shades; patient number's starting with IS; $n = 11$); **c** non-ICU survived (in green shades; patient number's starting with NS; $n =$ 6). Antibody off-rate constants that describe the fraction of antibody–antigen complexes decaying per second were determined directly from the plasma sample interaction with SARS-CoV-2 prefusion spike protein using SPR in the dissociation phase as described in Materials and Methods. All SPR experiments were performed twice blindly. The variation for each sample in duplicate SPR runs was <5%. The data shown is average value of two experimental runs. Off-rate was calculated and shown only for the samples that demonstrated a measurable (>5 RU) antibody binding in SPR. **d** The average antibody affinity against SARS-CoV-2 prefusion spike is shown for the final day samples from the COVID-19 patients who expired (red; $n = 8$ biologically independent individuals) vs. ICU-survived (blue; $n = 11$ biologically independent individuals) vs. non-ICU survived (green; $n = 6$ biologically independent individuals). Bar chart shows datapoints for each individual and presented as mean values ± SEM. **e** Fold-change in antibody affinity against SARS-CoV-2 prefusion spike was calculated for the final day samples compared with the first day sample from each of the COVID-19 patients who expired (red; $n = 8$ biologically independent individuals) vs. ICU-survived (blue; $n = 11$ biologically independent individuals) vs. non-ICU survived (green; $n = 6$ biologically independent individuals). Bar chart shows datapoints for each individual and presented as mean values ± SEM. The statistical significances between the groups were determined by non-parametric (Kruskal–Wallis) statistical test using Dunn's multiple comparisons analysis in GraphPad prism. The differences were considered statistically significant with a 95% confidence interval when the $p$-value was <0.05.

and macrophage activation syndrome[31–33]. Thus, the ability of virus-specific B cells to enter GC in lymph nodes and undergo affinity maturation may play an important role in the ultimate effectiveness of viral control and disease resolution in multiple viral infections.

The limitation of the current study is the lack of viral load measurements, which were not conducted on the hospitalized patients during the peak of pandemic. All patients' samples (>7 days post-symptom onset) in our study were SARS-CoV-2 negative for the virus, irrespective of disease outcome, therefore viral titer were not determined in this study. Nevertheless, our data underscores the realization that the immune response to SARS-CoV-2 infection is multifaceted and there may be several indicators of disease progression and outcome. In our study, sustained high levels of proinflammatory cytokines (IL-6 and IL-8), high serum IgA, and blunted affinity maturation against the prefusion spike protein were predictive of the worse outcome for the hospitalized patients. An elevated inflammatory response may be augmented by low-affinity antibodies that are not efficient in controlling SARS-CoV-2 replication. While disease resolution in survivors was associated with antibody affinity maturation to native prefusion spike demonstrating the importance of prefusion spike-specific antibodies as a predictive marker of COVID-19 disease outcome. The potential role of antibodies to native prefusion form of spike protein in protection from COVID-19 provides the impetus to measure antibody affinity against the SARS-CoV-2 prefusion spike in ongoing and planned therapeutic and vaccine studies against COVID-19.

In summary, our study highlights the need to perform comprehensive longitudinal analysis of immune response generated following SARS-CoV-2 infection to identify antibody biomarkers associated with protective immunity vs. immune failure that may contribute to the clinical disease outcome in COVID-19 patients. The quality of antibody specificity, isotype and affinity maturation provides an important insight that contributes to severity or resolution of COVID-19 disease that could inform development and evaluation of immune-based countermeasures against COVID-19.

## Methods

**Ethics statement.** The study protocol was approved by Center for Biologics Evaluation and Research (CBER), Food and Drug Administration (FDA) and performed under study protocol #CBER-2020-04-09 on de-identified plasma donations obtained from COVID-19 patients at the Adventist HealthCare White Oak and Shady Grove Medical Center. This study complied with all relevant ethical regulations for work with human participants, and informed consent was obtained. All adults hospitalized with COVID-19 disease were eligible without any specific selection criteria. Samples were collected from patients who provided informed consent to participate in the study. All assays performed fell within the permissible usages in the original informed consent.

**Plasma samples.** Heat inactivated plasma samples were obtained from Washington Adventist Medical Health Care and Shady Grove Adventist Hospital through Quest Diagnostics (Source Data). The plasma samples were collected serially from the day of admission until the patients were discharged or expired. All samples were deidentified. Initially, no patient information was provided, and all the immune analyses were conducted blindly. Subsequent patient information was provided and is presented in Source Data for the patients evaluated in the current study. This study was approved Food and Drug Administration's Research Involving Human Subjects Committee (FDA-RIHSC) (RIHSC #2020-04-02).

Surface plasmon resonance (SPR) measures antibody binding and antibody affinity to assess immune markers of protection mediated by the antibody binding and off-rate constants and requires multiple samples per participant. Assuming similar variability in change in antibody binding and antibody affinity over time, we calculated the power to detect differences in binding/affinity by group over time, comparing expired patients vs. ICU-survivors or non-ICU survivors. Power analysis calculations were done assuming a power value as 0.95, 0.9, and 0.8, in the order of decreasing stringency. A significance level of 0.05 was used. The SPR determined antibody binding or antibody affinity (off-rates) to prefusion SARS-CoV-2 spike protein for each group were used for sample size calculations. These calculations showed that we needed a sample size of 5.4, 4.3, and 3.1, respectively, that are within the actual sample size used in the current study. If antibody kinetics within individual is strongly associated over time, we will have higher power to detect this change.

**Proteins and monoclonal antibodies.** The SARS-CoV-2 Spike plasmid expressing genetically stabilized pre-fusion 2019-nCoV_S-2P *spike* ectodomain, a gene encoding residues 1-1208 of 2019-nCoV S fused to HisTag was a kind gift from Barney Graham (VRC, NIH). This expression vector was used to transiently transfect FreeStyle293-F cells (ThermoFisher) using polyethylenimine. Protein was purified from filtered cell supernatants using StrepTactin resin and subjected to additional purification by size-exclusion chromatography in phosphate-buffered saline (PBS).

**Measurement of cytokine levels in plasma.** All plasma samples were diluted 4-fold in Bio-Plex Sample Diluent HB buffer. The plasma samples were analyzed via a Bio-Plex Pro Human Cytokine Panel 17-Plex assay per the manufacturer's instructions. Plates were read using the Bio-Plex 200 system (Bio-Rad, Hercules, CA).

**SARS-CoV-2 pseudovirus production and neutralization assay.** Human codon-optimized cDNA encoding *SARS-CoV-2 S glycoprotein* (NC_045512) was synthesized by GenScript and cloned into eukaryotic cell expression vector pcDNA 3.1 between the BamHI and XhoI sites. Pseudovirions were produced by co-transfection Lenti-X 293T cells with pMLV-gag-pol, pFBluc, and pcDNA 3.1 SARS-CoV-2 S using Lipofectamine 3000. The supernatants were harvested at 48 h post transfection and filtered through 0.45-mm membranes.

For neutralization assay, 50 μL of SARS-CoV-2 S pseudovirions were pre-incubated with an equal volume of medium containing plasma at varying dilutions at room temperature for 1 h, then virus-antibody mixtures were added to Vero E6 cells in a 96-well plate. After a 12 h incubation, the inoculum was refreshed with fresh medium. Cells were lysed 48 h later, and luciferase activity was measured using luciferin-containing substrate.

**ACE2-RBD-binding ELISA.** Ninety-six-well Immulon plates were coated with recombinant SARS-CoV-2 RBD protein from 293T cells (RBD) in PBS overnight at 4 °C. After blocking, 100-fold dilution of plasma samples were added to the protein-coated plate in 100 μL for 1 h at ambient temperature. Plasma samples were assayed in duplicate. Naive plasma samples were assayed along with the experimental samples. After three washes with PBS/0.05% Tween 20, recombinant

human ACE2-AviTag (Acro biosystems; 40 ng/100 μL/well) followed by detection with an HRP-conjugated streptavidin (Jackson Immuno Research). After 1 h, plates were washed as before and OPD was added for 10 min. Absorbance was measured at 492 nm. Percent inhibition of ACE2 binding by the plasma was calculated based on the ACE2 binding in the absence of any plasma.

**Antibody binding kinetics of post-SARS-CoV-2 infection human plasma to recombinant SARS-CoV-2 prefusion spike protein by surface plasmon resonance (SPR).** Steady-state equilibrium binding of post-SARS-CoV-2 infected human polyclonal plasma was monitored at 25 °C using a ProteOn surface plasmon resonance (BioRad). The purified recombinant SARS-CoV-2 prefusion spike protein was captured via a His-tag to a Ni-NTA sensor chip with 200 resonance units (RU) in the test flow channels. The protein density on the chip was optimized such as to measure monovalent interactions independent of the antibody isotype[20]. Serial dilutions (10-, 50-, and 250-fold) of freshly prepared plasma in BSA-PBST buffer (PBS pH 7.4 buffer with Tween-20 and BSA) were injected at a flow rate of 50 μL/min (120 s contact duration) for association, and disassociation was performed over a 600-s interval. Responses from the protein surface were corrected for the response from a mock surface and for responses from a buffer-only injection. SPR was performed with serially diluted plasma of each individual time point in this study. Antibody isotype analysis for the SARS-CoV-2 spike protein bound antibodies in the polyclonal plasma was performed using SPR. Total antibody binding and antibody isotype analysis were calculated with BioRad ProteOn manager software (version 3.1). All SPR experiments were performed twice, and the researchers performing the assay were blinded to sample identity. Under these optimized SPR conditions, the variation for each sample in duplicate SPR runs was <5%. The maximum resonance units (Max RU) data shown in the figures was the calculated RU signal for the 10-fold diluted plasma sample.

Antibody off-rate constants, which describe the stability of the antigen-antibody complex, i.e., the fraction of complexes that decays per second in the dissociation phase, were determined directly from the human polyclonal plasma sample interaction with recombinant purified SARS CoV-2 prefusion spike ectodomain using SPR in the dissociation phase only for the sensorgrams with Max RU in the range of 10–100 RU and calculated using the BioRad ProteOn manager software for the heterogeneous sample model as described before[15,34]. Off-rate constants were determined from two independent SPR runs.

**SARS-CoV-2 gene fragment phage display library (GFPDL) construction.** DNA encoding the *spike* gene of SARS-CoV-2 isolate Wuhan-Hu-1 strain (GenBank: MN908947.3) was chemically synthesized and used for cloning. A gIII display-based phage vector, fSK-9-3, was used where the desired polypeptide can be displayed on the surface of the phage as a gIII-fusion protein. Purified DNA containing *spike* gene was digested with *DNase*I to obtain gene fragments of 50–1500 bp size range and used for GFPDL construction[17,20]. The phage libraries were constructed from the SARS-CoV-2 *spike* gene potentially display viral protein segments ranging in size from 18 to 500 amino acids, as fusion protein on the surface of bacteriophage.

**Affinity selection of SARS-CoV-2 GFPDL phages with polyclonal human plasma.** Prior to panning of GFPDL with polyclonal plasma antibodies, plasma components that could non-specifically interact with phage proteins were removed by incubation with UV-killed M13K07 phage-coated Petri dishes[17]. Equal volumes of each human plasma were used for GFPDL panning. GFPDL affinity selection was carried out in-solution with anti-IgM, or protein A/G (IgG), or anti-IgA-specific affinity resin[17,20]. Briefly, the individual plasma was incubated with the GFPDL and the specific resin, the unbound phages were removed by PBST (PBS containing 0.1% Tween-20) wash followed by washes with PBS. Bound phages were eluted by addition of 0.1 N Gly-HCl, pH 2.2 and neutralized by adding 8 μL of 2 M Tris solution per 100 μL eluate. After panning, antibody-bound phage clones were amplified, the inserts were sequenced, and the sequences were aligned to the SARS-CoV-2 *spike* gene, to define the fine epitope specificity in these COVID-19 patients using MacVector version 17.5.2.

The GFPDL affinity selection data were performed in duplicate (two independent experiments by research fellow in the lab, who was blinded to sample identity). Similar numbers of bound phage clones and epitope repertoire were observed in the two GFPDL panning.

**Adsorption of polyclonal human post-infection plasma on SARS-CoV-2 GFPDL and residual reactivity to SARS-CoV-2 prefusion spike.** Prior to panning of GFPDL, 500 μl of 10-fold diluted plasma pooled from post-infection samples from either expired individual or non-ICU survivor were incubated with SARS-CoV-2 GFPDL coated petri dishes. To ascertain the residual antibodies specificity, an ELISA was performed with wells coated with 100 ng/100 μl of purified recombinant SARS-CoV-2 prefusion spike. After blocking with PBST containing 2% BSA, serial dilutions of human plasma (with or without adsorption) in blocking solution were added to each well, incubated for 1 h at room temperature (RT), followed by addition of 5000-fold diluted HRP-conjugated goat anti-human IgA + IgG + IgM-specific antibody and developed by 100 μl of OPD substrate solution. Absorbance was measured at 490 nm.

**Peptide ELISA.** Immulon 2 HB 96-well microtiter plates were coated with 100 μl of streptavidin in PBS (100 ng/well) at 4 °C overnight followed by capturing of biotinylated SARS-CoV-2 synthetic peptides. After blocking with PBST containing 2% BSA, five-fold serial dilutions of post-infection human serum in blocking solution were added to each well, incubated for 1 h at RT, followed by addition of 2000-fold dilution of HRP-conjugated goat anti-human IgG-Fc specific antibody, and developed by 100 μL of OPD substrate solution. Absorbance was measured at 490 nm.

**Statistical analysis.** Statistical differences were performed using GraphPad prim version 8 (Graph Pad software Inc, San Diego, CA). The statistical significances between the groups were determined by non-parametric (Kruskal–Wallis) statistical test using Dunn's multiple comparisons analysis in GraphPad prism The differences were considered statistically significant with a 95% confidence interval when the *p*-value was <0.05. For trendline fit, a non-linear regression model was used and the trendline was fit by polynomial fourth order through the origin. The method was chosen by least squares regression and the best fit values of selected unshared parameters between data sets. Confidence intervals (CI) of parameters were chosen to be 95% asymmetrical CI.

**Reporting summary.** Further information on research design is available in the Nature Research Reporting Summary linked to this article.

## Data availability
All data are shown in the manuscript figures and supplementary information. Antigenic sites were depicted on the SARS-CoV-2 spike structure PDB#6VSB (https://www.rcsb.org/structure/6VSB). Sequence for SARS CoV-2 spike protein (Genbank#MN908947), SARS CoV-1 BJ01 strain (Genbank#AAP30030.1), MERS CoV KOR/KNIH/2015 (Genbank#AKN11075.1), Bat SARS-like CoV ZC45 (Genbank#AVP78031.1), Human CoV NL63 (NCBI#YP_003767.1), and Human CoV HKU1 (NCBI#YP_173238.1) were downloaded from https://www.ncbi.nlm.nih.gov/genbank/. Source data are provided with this paper.

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

## Acknowledgements
We thank Keith Peden and Basil Golding for their insightful review of the manuscript. We thank James Rost and Norton Elson of Washington Adventist Medical Health Care, Nicolas Cacciabeve of Advanced Pathology Associates, and Rob San Luis, Maryonne Snow-Smith, Demetra Collier, Meaza Belay, Genevieve Caoili, Zanetta E. Morrow, Bryana Streets of Quest Diagnostics for collection of clinical samples and support of the clinical study. The research work described in this manuscript was supported by FDA intramural funds and NIH-NIAID IAA #AAI20040. The funders had no role in study design, data collection and analysis, decision to publish, or preparation of the manuscript. The content of this publication does not necessarily reflect the views or policies of the Department of Health and Human Services, nor does mention of trade names, commercial products, or organizations imply endorsement by the U.S. Government.

## Author contributions
Designed research: S.K. and H.G. Clinical specimens and unblinded clinical data: H.G. of Adventist Healthcare. Performed research: J.T., S.R., Y.L., E.C., L.K., G.G., and S.K. Contributed to writing: S.K. and H.G.

## Competing interests
The authors declare no competing interests.
