## [Peer Review File · Nature Communications]

REVIEWERS' COMMENTS

Reviewer #1 (Remarks to the Author):

I am generally satisfied with the authors responses. One request: please comment in the manuscript whether GFPDL using a RBD or other Spike Mabs for which the epitope is known yields the expected clones i.e epitope containing peptides.

Reviewer #2 (Remarks to the Author):

In this revised manuscript the authors have addressed all of the initial concerns that were raised with substantial new data and clarification to the writing. This work represents a significant contribution to our understanding of SARS-CoV-2 antibody responses.

Reviewer #3 (Remarks to the Author):

The review addressed some of the previous concerns raised but has not materially improved the impact of this study. It is now clear that viral load was either not tracked or not detectable at late times after infection, even in severe disease. The response to the critique that many of the antibody studies tracked potentially irrelevant linear epitopes was responded to with measurement of reactivity to—synthetic peptides. It was also made clearer that there is no statistically significant correlation between severe disease and neutralizing or ACE2 blocking antibodies. There seems to be no reliable way to prospectively use the antibody response information from this paper to predict patient outcomes.

Reviewer #1:

Reviewer #1 (Remarks to the Author):

I am generally satisfied with the authors responses. One request: please comment in the manuscript whether GFPDL using a RBD or other Spike Mabs for which the epitope is known yields the expected clones i.e epitope containing peptides.

Response:

We appreciate Reviewer' recommendation. During characterization of our GFPDL, we have earlier performed epitope mapping of some MAbs against Spike and RBD. We have added the following sentence in the text:

Lines 135-6:

GFPDL based epitope mapping of monoclonal antibodies (MAbs) targeting SARS-CoV-2 spike or RBD identified the expected epitope recognized by these MAb.

Reviewer #2 (Remarks to the Author):

In this revised manuscript the authors have addressed all of the initial concerns that were raised with substantial new data and clarification to the writing. This work represents a significant contribution to our understanding of SARS-CoV-2 antibody responses.

Response:

We appreciate the recommendation for publication in Nature Communications.

Reviewer #3 (Remarks to the Author):

The review addressed some of the previous concerns raised but has not materially improved the impact of this study. It is now clear that viral load was either not tracked or not detectable at late times after infection, even in severe disease. The response to the critique that many of the antibody studies tracked potentially irrelevant linear epitopes was responded to with measurement of reactivity to— synthetic peptides. It was also made clearer that there is no statistically significant correlation between severe disease and neutralizing or ACE2 blocking antibodies. There seems to be no reliable way to prospectively use the antibody response information from this paper to predict patient outcomes.

Response:

We agree with reviewer that binding to linear epitopes may be clinically irrelevant, as well as neutralizing titers/ACE2 inhibition do not correlate with disease outcome as mentioned in discussion section (lines 275-280). To identify an immune correlate, therefore, we focused on analyzing individual serum/plasma sample antibody kinetics using SARS-CoV-2 trimeric prefusion spike protein that is hypothesized to mimic native spike protein on SARS-CoV-2 virion and is a component of several vaccines in advanced clinical trials.

We have discussed these observations throughout the manuscript and expanded the discussion.

(lines 318-320): Our study identified antibody affinity maturation of prefusion spike-binding antibodies longitudinally during the hospital stay provided a clear antibody correlate differentiating between the survivors vs. fatal cases of COVID-19 disease.

(lines 350-360) : In our study, sustained high levels of proinflammatory cytokines (IL-6 and IL-8) high serum IgA, and blunted affinity maturation against the prefusion spike protein were predictive of the worse outcome for the hospitalized patients. While disease resolution in survivors was associated with antibody affinity maturation to native prefusion spike demonstrating the importance of prefusion spike specific antibodies as a predictive marker of COVID-19 disease outcome. The potential role of antibodies to native prefusion form of spike protein in protection from COVID-19 disease provides the impetus to measure antibody affinity against the SARS-CoV-2 prefusion spike in ongoing and planned therapeutic and vaccine studies against COVID-19.